# MDR: Multi-stage Decoupled Relational Knowledge Distillation with Adaptive Stage Selection

## ABSTRACT

The effectiveness of contrastive-learning-based Knowledge Distillation (KD) has sparked renewed interest in relational distillation, but these methods typically focus on angle-wise information from the penultimate layer. We show that exploiting relational information derived from intermediate layers further improves the effectiveness of distillation. We also find that adding distance-wise relational information to contrastive-learning-based methods negatively impacts distillation quality, revealing an implicit contention between angle-wise and distance-wise attributes. Therefore, we propose a **M**ulti-stage **D**ecoupled **R**elational (MDR) KD framework equipped with an adaptive stage selection to identify the stages that maximize the efficacy of transferring the relational knowledge. MDR framework decouples angle-wise and distance-wise information to resolve their conflicts while still preserving complete relational knowledge, thereby resulting in an elevated transferring efficiency and distillation quality. To evaluate the proposed method, we conduct extensive experiments on multiple image benchmarks (*i.e.* CIFAR100, ImageNet and Pascal VOC), covering various tasks (*i.e.* classification, few-shot learning, transfer learning and object detection). Our method exhibits superior performance under diverse scenarios, surpassing the state of the art by an average improvement of 1.22% on CIFAR-100 across extensively utilized teacher-student network pairs.

## CCS CONCEPTS

• **Computing methodologies** → **Computer vision**.

## KEYWORDS

relation-based knowledge distillation, multi-stage, decouple

## 1 INTRODUCTION

Over the past decades, unprecedented development in neural networks has created numerous multimedia applications on vision and/or language, ranging from image classification [8, 13, 20], object detection [25], and visual question answering [17]. However, these neural networks demand substantial computational and storage resources due to their large model sizes, resulting in expensive and cumbersome model deployment. To address this limitation, various model compression techniques have been systematically explored, such as pruning [10], quantization [7], Neural Architecture Search

(NAS) [33], and Knowledge Distillation (KD) [12]. Among them, KD stands out for its compatibility with other compression techniques, superior generalization ability [2, 19, 42] and model structure flexibility, thus making it vital in applications such as object detection [3, 36] and Multiple Object Tracking (MOT) [16, 41].

KD aims to transfer knowledge from a heavy-weight model (teacher) to a light-weight one (student). A straightforward approach is to align the student's output probability distribution with that of the teacher [12]. However, due to the limited scope of information in this distribution, subsequent research has shifted towards matching outputs of intermediate layers [11, 27], which further bifurcates into feature-based and relation-based methods. ReviewKD [2], a feature-based method, exploits the residual structure to selectively refine the outputs of multiple intermediate layers. In comparison, prominent relation-based approaches excel by combining the relational matrix from multiple samples with contrastive learning in unsupervised domain. Particularly, SSKD [34], a relation-based method, transfers knowledge by fitting the angle-wise relational matrix composed of positive and negative sample pairs.

Despite these successes, we argue that existing contrastive learning based distillation methods have yet to realize their full potential for two primary reasons. First, these methods only use *single-stage* output features for relationship extraction, thereby overlooking the utility of *multi-stage* relational information between samples. Second, they rely only on angle-wise information, neglecting the informative distance component for relational representation.

However, leveraging these missed opportunities presents certain challenges. On one hand, as shown in Fig.1a, the mere incorporation of multi-stage relational information does not necessarily improve the distillation efficacy, even when the volume of transferred information increases. This suggests that raw multi-stage relational data may introduce redundant or even harmful information during the knowledge transfer from the teacher to the student model. On the other hand, Fig. 1b highlights the limitations of solely relying on angle-wise relationships. We calculated the length distribution (denotes the distance from the origin point) of the penultimate layer's output from the student model, in order to eliminate the influence of angle-wise information. Fig. 1b shows the length distribution of penultimate features from models trained by various KD methods, where a larger overlapping area with the teacher's distribution implies greater retention of distance information. This observation reveals that using only the angle-wise relationship between samples for knowledge distillation leads to evident information loss of the length distribution. Moreover, as demonstrated by RKD [22], directly fitting both angle-wise and distance metrics between samples results in complex, interdependent matrices, and thereby culminates in sub-optimal performance.

To address these constraints, we introduce the Multi-stage Decoupled Relational (MDR) knowledge distillation framework. Utilizing a novel Adaptive Stage Selection (ADSS) strategy, MDR selects

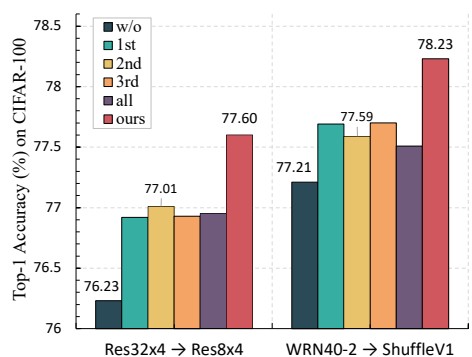
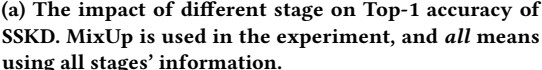

(a) The impact of different stage on Top-1 accuracy of SSKD. MixUp is used in the experiment, and *all* means using all stages' information.

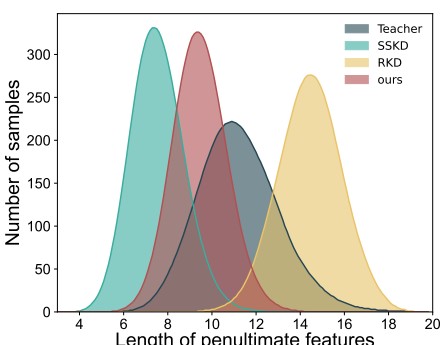

(b) Length distribution of individual sample's penultimate features from models trained by different KD methods (Res32×4 → Res8×4).

Figure 1: Experimental results on relational information of samples in CIFAR100.

the most suitable stages for each sample based on the relational representation capability of both its angle-wise and distance-wise relational information. In addition, MDR decouples inter-sample relationships into angle-wise and length-wise dimensions, allowing for a simultaneous and conflict-free transfer of both types of information. Moreover, in order to prevent Self-supervised Module (SM) from neglecting length information during feature normalization in contrastive-learning-based methods, we present a novel training methodology for SMs that replaces the training based on contrastive learning with an auxiliary classifier. This modification preserves length attributes while enhancing angle-wise representation capability. Our evaluation demonstrates that MDR framework surpasses the state of the art (SOTA) by an average of 1.22% on CIFAR-100 across extensively utilized network pairs. In summary, this paper makes the following contributions:

- We present critical insights into the constraints of the existing contrastive learning based knowledge distillation frameworks. We delineate the avenues to further improve the methods through the optimized selection of multi-stage information and the strategic decoupling of angle-wise and length-wise relational representation.
- We propose adaptive stage selection to enable multi-stage information extraction. We also present the concept of relationship decoupling to partition relationships into angle-wise and length-wise components for a streamlined student training process. Moreover, we formulate a new SM training paradigm to compensate for the loss of length-wise information and augment its contrastive learning representation capability.
- We cohesively integrate these innovations into a novel MDR distillation framework. Our comprehensive evaluation results show that MDR framework consistently exceeds SOTA performance across extensively utilized network pairs on CIFAR-100, with an accuracy improvement up to 1.22%.

## 2 RELATED WORK

Knowledge distillation trains a smaller network using the knowledge from a larger network. Based on the types of the knowledge, existing KD frameworks can be divided into three categories: *response-based*, *feature-based* and *relation-based* methods [6].

**Response-based** KD, also known as the classic KD [12], usually relies on the neural response of the last output layer of the teacher model. The main idea is to directly mimic the final prediction (logits) of the teacher model. DKD [42] proposes a decoupled approach using the fundamental concept of KD. Unlike our proposed decoupled approach, DKD only separates the output categories into target and non-target classes, and assigns different importance to them. HSAKD [35] trains separate classifiers for each stage while transferring multi-stage response-based information.

**Featured-based** KD, represented by FitNet [27], encourages the student models to mimic the intermediate-level features from the hidden layers of teacher models. VID [1] and PKT [23] reformulate knowledge distillation as a procedure of maximizing the mutual information between the teacher and the student networks. There are also other methods using multi-stage information to transfer knowledge. OFD [11] uses a novel distance function to transfer knowledge from teacher to student; ReviewKD [2] proposes a new multi-stage architecture that allows the student to select the appropriate teacher stage for distillation. In contrast, our method employs an adaptive stage selection strategy to extract the most relevant relational information applicable to distillation for different samples.

**Relation-based** KD emphasizes the exploitation of relationships between distinct layers or samples. FSP [37] guides the student model by generating a relation matrix between different layers of the teacher model. SP [32], CC [24], and RKD [22] utilize the relationships between samples to guide the student in learning higher-dimensional representations. Leveraging the success of contrastive learning in unsupervised tasks [22, 24], many methods utilize the representation space of contrastive learning to model

the relationships between samples. CRD [31] pioneered the integration of contrastive learning into knowledge distillation, SSKD [34] separately trains the teacher's SM to extract richer knowledge, PACKD [38] uses an optimal transport-based positive pair similarity weighting strategy to better transfer discriminative information from teachers to students. However, all of the existing contrastive-learning-based methods extract relationships at the penultimate feature layer. According to our experiments, we found that effective relational information can also be extracted from intermediate layers. Therefore, we propose a multi-stage distillation framework with adaptive stage selection strategy to comprehensively extract relational knowledge between samples. Moreover, our novel method decouples the relationship between samples into angle and length difference, compensating information loss in length-wise relationships in the existing contrastive-learning-based methods.

## 3 METHODOLOGY

In this section, we first provide a brief review of KD and the details of contrastive-learning-based knowledge distillation methods. In light of the aforementioned problems and limitations of the existing methods, we present our proposed framework featuring an adaptive stage selection strategy, followed by the concept of relationship decoupling.

### 3.1 Preliminary

The *response-based* methods transfer the *dark knowledge* from the teacher by approximating the distribution of soft targets, which can be formulated as:

$$\mathcal{L}_{kd} = \tau^2 KL(\sigma(z^s; \tau) \| \sigma(z^t; \tau)), \qquad (1)$$

where $z^s$ and $z^t$ are the logits from the student and the teacher respectively; $\sigma(\cdot)$ is the softmax function that produces the category probabilities from the logits, and $\tau$ is a temperature hyperparameter to scale the smoothness of the distribution; $KL$ means Kullback-Leibler divergence, which is the measurement of dissimilarity between two categorical distributions.

The main idea of the *feature-based* KD methods is to mimic the feature representations between student and teacher, which can be formulated as the following loss function:

$$\mathcal{L}_{feat} = \sum_k \mathcal{L}_f(\mathcal{T}_s(F_k^s), \mathcal{T}_t(F_k^t)), \qquad (2)$$

where for stage $k$, $F_k^s$ and $F_k^t$ denote the feature maps from the student and the teacher respectively; $\mathcal{T}_s, \mathcal{T}_t$ denote the student and the teacher transformation module respectively; $\mathcal{L}_f(\cdot)$ denotes the function which compute the distance between two feature maps. Using multi-stage information has become the prevailing approach for feature-based methods [2, 35].

In contrast to the methods that distill knowledge from individual samples, the *relation-based* KD methods exploit the relationship between distinct samples, which can be formulated as:

$$\mathcal{L}_{rela}(F_t, F_s) = \mathcal{L}_{R^2}(\psi(t_i, t_j), \psi(s_i, s_j)), \qquad (3)$$

where $(t_i, t_j) \in F_t$ and $(s_i, s_j) \in F_s$, $F_t$ and $F_s$ are the sets of feature representations of samples from the teacher and student respectively; $\psi(\cdot)$ denotes the similarity function of $(t_i, t_j)$ or $(s_i, s_j)$;

$\mathcal{L}_{R^2}(\cdot)$ is the correlation function of the feature representations between teacher and student (*e.g.*, Huber loss). However, the existing relation-based methods focus on the design of the relational matrix and neglect the valuable multi-stage information.

As the predominant *relation-based* method, contrastive-learning-based knowledge distillation captures inter-sample relationships to transfer knowledge by leveraging the cosine similarity within the representation space. Given a mini-batch containing $N$ samples $\{x_i\}_{i=1:N}$ (*i.e.*, anchor set $\mathcal{P}$), we apply strong data augmentation $t(\cdot)$, such as Random Rotation [34] or MixUp [38], to each sample and obtain $\{\widetilde{x}_i\}_{i=1:M}$ ((*i.e.*, positive set $\widetilde{\mathcal{P}}$) where $M = 3N$. Both $x_i$ and $\widetilde{x}_i$ are fed into the teacher or student networks to extract their representations $\phi_i = f(x_i), \widetilde{\phi}_i = f(\widetilde{x}_i)$. The similarities between $x_i$ and $\widetilde{x}_i$ can be represented by the following matrix $\mathcal{A}$:

$$\mathcal{A}_{i,j} = cosine(\widetilde{z}_i, z_j) = \frac{dot(\widetilde{z}_i, z_j)}{\|\widetilde{z}_i\|_2 \|z_j\|_2}, \qquad (4)$$

where $\widetilde{z}_i$ and $z_j$ are the outputs of SM, which transforms $\widetilde{\phi}_i$ and $\phi_i$ into a contrastive learning representation space. $\mathcal{A}_{i,j}$ represents the similarity between $\widetilde{x}_i$ and $x_j$. $(\widetilde{x}_i, x_i)$ refers to the positive pair and $(\widetilde{x}_i, x_j)_{i \neq j}$ the negative pair. The SM consists of a 2-layer perceptron with a pooling layer, which is trained by maximizing the similarity between positive pairs. A commonly used contrastive objective is defined as:

$$\mathcal{L}_{con} = -\sum_i log \frac{\exp(\mathcal{A}_{i,i}/\tau)}{\sum_k \exp(\mathcal{A}_{i,k}/\tau)}. \qquad (5)$$

In addition to the angle-wise relational matrix formation, the distance-wise relational matrix between samples can also be used to transfer knowledge [22], which can be expressed as:

$$\mathcal{D}_{i,j} = \|\widetilde{z}_i - z_j\|_2. \qquad (6)$$

However, utilizing both the angle-wise and distance-wise matrices simultaneously leads to a degraded performance due to their strongly coupled relationship.

### 3.2 Adaptive Stage Selection Strategy

As mentioned in Sec.1, the existing contrastive-learning-based knowledge distillation methods only use single-stage output features for relationship extraction. However, as shown in Fig. 1a, the output of each stage contains valuable angle-wise relational information for the student to learn. To better exploit these information, we adopt a multi-stage framework to transfer knowledge. Specifically, for each distillation stage, both teacher and student networks are equipped with SMs to capture relational information. We augment the data set using MixUp and derive the anchor set $\mathcal{P}$ and the positive set $\widetilde{\mathcal{P}}$. To fully exploit the representation capability, we incorporate both the angle-wise and distance-wise information instead of solely relying on angle-wise relationships in the loss function. Therefore, unlike Eqn. 3, the loss of multi-stage knowledge transfer is represented as:

$$\mathcal{L}_{rela} = \sum_k \sum_{i \in \widetilde{\mathcal{P}}, j \in \mathcal{P}} \mathcal{L}_{R^2}(\mathcal{B}_{i,j}^{s,k} \| \mathcal{B}_{i,j}^{t,k}), \qquad (7)$$

where for $k$-th stage, $\mathcal{B}^s$ is a probability matrix, consisting of student's similarity matrix $\mathcal{A}^s$ (Eqn. 4) or $\mathcal{D}^s$ (Eqn. 6) with softmax (with temperature scale $\tau$) along the dimension of all samples from

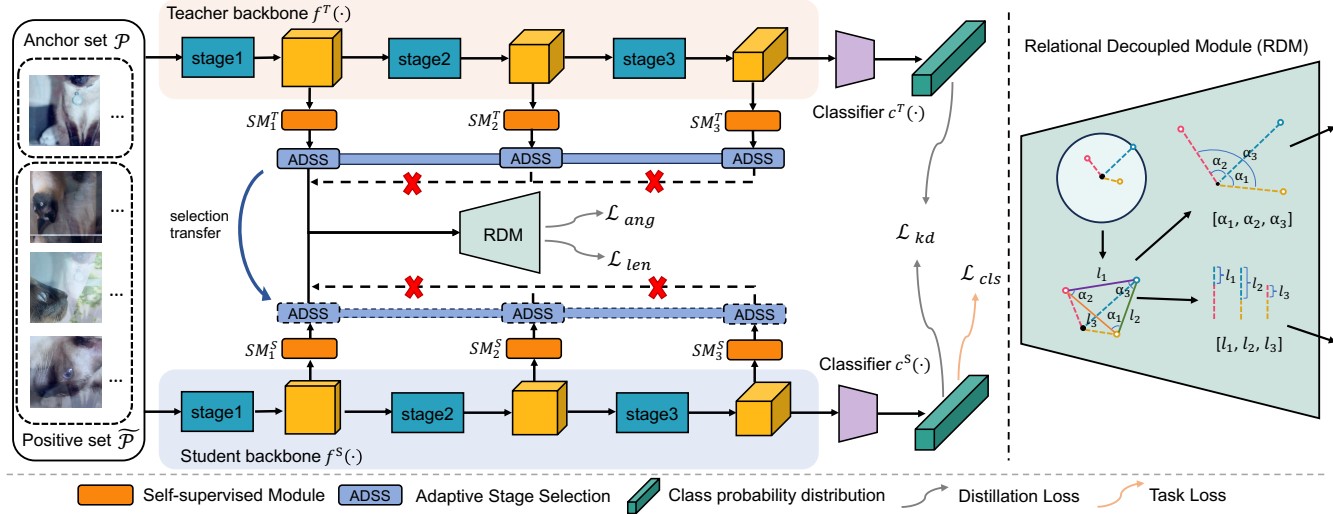

**Figure 2: Illustration of our proposed MDR framework. ADSS selects the appropriate stage for information transfer (the first stage is selected in this example). Relational Decoupled Module (RDM) transforms the relational information among multiple samples within the corresponding stage into angle-wise (*e.g.* $[\alpha_1, \alpha_2, \alpha_3]$) and length-wise (*e.g.* $[l_1, l_2, l_3]$) representations. $\mathcal{L}_{ang}$ and $\mathcal{L}_{len}$ are used to transfer decoupled relational information. The red cross indicates that the information at this stage is filtered in this case.**

$\mathcal{P}$ in the mini-batch. The same procedure is applied to the teacher to obtain $\mathcal{B}^t$.

As shown in Fig. 1a, incorporating multi-stage relational information straightforwardly does not necessarily improve the distillation efficacy. We argue that every stage contains beneficial information, but using all stages introduce redundant or even harmful information during the knowledge transfer. In order to obtain just enough relational information effectively, we propose an adaptive stage selection strategy to select the most appropriate stage for angle-wise knowledge transfer.

Since both the intra-class (positive pairs) and inter-class (negative pairs) correlation can reflect the representational capability, it is insufficient to use only the cosine similarity of positive pairs as a metric. Moreover, the representation capabilities of each stage are different, and it is not suitable to directly compare the absolute values of similarity between positive and negative pairs. Therefore, we use relative numerical ranking instead of absolute cosine similarity. Specifically, we use Eqn. 4 to calculate the cosine similarity between each positive anchor pair and rank them individually at each stage of the mini-batch extending the dimensions of the positive sample. By comparing the order of the similarity between a positive sample and its corresponding anchor in each stage, we select the highest-ranking stage for knowledge transfer.

As illustrated in Fig. 2, we exploit distance-wise relationship, in order to convey more comprehensive information during distillation. As for the stage selection, we use absolute distance as the criterion and also exploit relative numerical ranking by using the following formula:

$$AS(\{M_i^k\}_{k=1}^K) = \arg\min_k \{Rank(M_i^k)\}_{k=1}^K, \tag{8}$$

where $K$ is the number of stages in a network; $M_i^k$ is angle-wise or distance-wise similarity matrix between positive sample $i$ and all anchor samples in the mini-batch; $Rank$ is the function that sorts the similarity in descending order.

### 3.3 Relational Decoupled Module

As shown in Fig. 1b, exclusively depending on angle-wise relationship during distillation leads to information loss of the length distribution. Therefore, it is crucial to utilize both angle-wise and distance-wise information in order to comprehensively capture the relational information between samples for distillation. Yet, it is challenging to effectively combine these two types of information. For example, RKD directly used two types of information, but the best distillation results are often obtained by taking one of the two. This is because the distance metric contains both angle and length (the latter indicating distance from the origin) information, which may obstruct the comprehension of angle information while learning distance information. In the feature representation space, the sample distance does not align with the principles of contrastive learning as described in Eqn. 5. Specifically, the distance between the positive pairs does not always need to be close. Therefore, conflicts arise when fitting angle-wise and distance-wise relationships simultaneously.

To solve this problem, we propose the concept of relationship decoupling. As illustrated in Fig. 2, we decouple the relationship between samples into angle and length difference, and the latter can be expressed by the following equation:

$$\mathcal{D}iff_{i,j} = \frac{1}{\mu_i} \left| \, ||\widetilde{z_i}||_2 - ||z_j||_2 \, \right|, \tag{9}$$

where $\mu$ is a normalization factor for length difference. Similar to RKD, we set $\mu$ to be the average length difference between pairs from $\mathcal{P}$ and $\widetilde{\mathcal{P}}$ in the mini-batch:

$$\mu_i = \frac{1}{|\mathcal{P}^2|} \sum_{j \in \mathcal{P}} \big| \, ||\widetilde{z_i}||_2 - ||z_j||_2 \, \big|. \tag{10}$$

Unlike the traditional way of transferring angle-wise knowledge, we directly use MSE loss to fit the length-wise relational matrix. The length-wise loss and angle-wise loss are defined respectively as:

$$\mathcal{L}_{len} = \sum_{i \in \widetilde{\mathcal{P}}, j \in \mathcal{P}} MSE(\mathcal{D}iff_{i,j}^{s,k}, \mathcal{D}iff_{i,j}^{t,k}) \tag{11}$$

$$\text{s.t.} \quad k = AS(\{\mathcal{D}_{i,j}^{t,k}\}_{k=1}^{K}),$$

$$\mathcal{L}_{ang} = \tau^2 \sum_{i \in \widetilde{\mathcal{P}}, j \in \mathcal{P}} KL(\mathcal{B}_{i,j}^{s,k} \| \mathcal{B}_{i,j}^{t,k}) \tag{12}$$

$$\text{s.t.} \quad k = AS(\{\mathcal{B}_{i,j}^{t,k}\}_{k=1}^{K}).$$

The final loss for the student network is the combination of aforementioned terms, including the original training loss $\mathcal{L}_{cls}$ , the response-based loss $\mathcal{L}_{kd}$, and the relation-based loss $\mathcal{L}_{ang}$ and $\mathcal{L}_{len}$:

$$\mathcal{L} = \lambda_1 \mathcal{L}_{cls} + \lambda_2 \mathcal{L}_{kd} + \lambda_3 \mathcal{L}_{ang} + \lambda_4 \mathcal{L}_{len}, \tag{13}$$

where the $\lambda_i$ is the balancing weight.

Before training the student, we freeze the teacher's backbone and train the SM. SM is typically trained by explicitly improving the representational ability of contrastive learning (Eqn. 5) in existing methods, which neglect length-wise information during feature normalization to prioritize angle-wise relationships. To preserve length-wise information while maintaining the representational ability of contrastive learning, we place a classifier behind each SM and directly use cross-entropy (CE) loss. This approach ensures that the dimensions of the outputs are consistent while retaining the relational information. Compared with the prior training methods, the contrastive learning representational ability of SM is further amplified with CE loss. Moreover, training through a classifier is more effective for SM to obtain the global information of the data set, rather than the relational information between samples within a mini-batch.

## 4 EXPERIMENTS

To demonstrate the effectiveness of our work, we evaluate MDR in various tasks: *classification*, *few-shot learning*, *transfer learning* and *object detection*. Moreover, we present various ablation study for the proposed method.

### 4.1 Experimental Settings

**Datasets and Competitors** We conduct evaluations on standard CIFAR-100 [15] and ImageNet [28] benchmarks across the widely applied network families including ResNet [9], WRN [40], VGG [30], MobileNet [29], ShuffleNet [21]. CIFAR-100 [15] contains 50K images for training and 10K images for testing, labeled into 100 fine-grained categories. The size of each image is 32×32. ImageNet [28] consists of 1.2M images for training and 50K images for validation, covering 1,000 categories. All images are resized to 224 × 224 during training and testing. We report the top-1 and top-5 accuracy on this dataset for image recognition.

Moreover, we employ the SIL-10 [4] and TinyImagenet [28] datasets to assess the transferability of learned representations generated by distillation method. STL-10 [4] is composed of 5K labeled training images and 8K test images in 10 classes. TinyImageNet [28] is composed of 100K training images and 10k test images in 200 classes. We evaluate the proposed MDR on this dataset with image recognition and report the top-1 accuracy.

Following the consistent protocol, we use Pascal VOC [5] train-val07 + 12 for training and test07 for evaluation. The result set consists of 16551 training images and 4952 test images in 20 classes. The image scale is $1000 \times 600$ pixels during training and inference. The comparison of detection performance toward average precision (AP) on individual classes and mean AP (mAP).

We compare MDR with a wide range of representative KD methods, including KD [12], FitNets [27], AT [39], SP [32], CC [24], RKD [22], PKT [23], OFD [11], CRD [31], SSKD [34], CRCD [43], ReviewKD [2], DKD [42], CTKD [18], ML-LD [14].

**Implementation details** We attach one SM after each convolutional stage. The SM is composed of global average pooling(GAP) and two fully-connected(FC) layer. For training teacher SMs, we attach one FC layer for CE loss, where the input dimension is same as the dimension of SM's output feature (*e.g.*, 128 on CIFAR-100, 1280 on ImageNet) and the output dimension is same as the number of categories. During the training stage of teacher's SMs, we connect a classifier after each SM (composed of a layer of fully connected), and directly uses category information for supervised learning. In this process, except for SM and classifier, the backbone part of the network remains frozen.

On CIFAR-100, the batch size and initial learning rate are set to 64 and 0.05. We train the models for 240 epochs in total with SGD optimizer, and decay the learning rate by 0.1 at 150, 180, and 210 epochs. The weight decay and the momentum are set to 5e-4 and 0.9. On ImageNet, we adopt the SGD optimizer to train the student networks for 100 epochs with a batch size of 512. The initial learning rate is 0.2 and decayed by 10 when the epoch is 30, 60 and 90. Weight decay and momentum are the same as above. We set $\tau$ in $\mathcal{L}_{kd}$ for $\mathcal{P}$ to be 1, $\widetilde{\mathcal{P}}$ to be 1, $\tau$ in $\mathcal{L}_{ang}$ and $\mathcal{L}_{len}$ to be 0.5. We set $\lambda_1 = 1.0, \lambda_2 = 2.0, \lambda_3 = 300, \lambda_4 = 1.0$ in Eqn. 13.

Due to the page limit, we provide more training details in the supplementary materials.

### 4.2 Comparison with the State Of The Arts

**Results on CIFAR-100.** We compare our MDR with representative distillation methods using a variety of teacher-student pairs, with both identical and different architectural styles on the CIFAR-100 dataset. As shown in Table 1, our MDR consistently outperforms other methods by a significant margin. Specifically, our method achieves an average of 0.88% accuracy improvement when compared with the existing optimal method for each network pair configuration. The amount of accuracy improvement is larger than that of many previous methods. Collectively, there's an average improvement of 1.22% in accuracy over the best-performing SSKD (more experiments in the supplementary materials). These results indicate that our proposed MDR effectively exploits the decoupled

**Table 1: Top-1 accuracy (%) comparison of SOTA distillation methods across various teacher-student pairs on CIFAR-100. The numbers in Bold and underline denote the best and the second-best results, respectively.**

| Teacher
Acc.
Student
Acc. | WRN40-2
76.41
WRN40-1
71.98 | WRN40-2
76.41
WRN16-2
73.26 | ResNet56
73.44
ResNet20
69.06 | ResNet110
74.07
ResNet32
71.45 | VGG13
75.38
VGG8
70.68 | ResNet32×4
79.42
ResNet8×4
72.50 | ResNet32×4
79.42
ShuffleV2
71.82 | ResNet50
79.34
MobileV2
64.60 | Avg |
|---|---|---|---|---|---|---|---|---|---|
| KD | 73.99 | 75.81 | 71.31 | 73.23 | 73.33 | 73.69 | 74.73 | 68.09 | 73.02 |
| FitNet | 74.44 | 75.63 | 71.59 | 73.26 | 74.02 | 75.28 | 75.30 | 66.77 | 73.29 |
| AT | 74.67 | 75.77 | 71.60 | 74.03 | 73.92 | 75.42 | 75.51 | 67.20 | 73.52 |
| SP | 73.91 | 75.44 | 71.02 | 73.88 | 73.31 | 74.09 | 75.20 | 69.11 | 73.25 |
| CC | 73.98 | 75.41 | 71.43 | 74.30 | 73.39 | 74.87 | 75.44 | 69.34 | 73.52 |
| RKD | 73.91 | 75.33 | 70.74 | 73.54 | 73.66 | 74.85 | 75.50 | 68.82 | 73.29 |
| PKT | 74.78 | 75.42 | 71.78 | 73.99 | 73.65 | 74.45 | 76.00 | 68.72 | 73.60 |
| CRD | 74.45 | 75.89 | 71.55 | 74.24 | 74.08 | 75.88 | 76.46 | 69.76 | 74.04 |
| CRCD | 74.41 | 76.07 | 71.49 | 73.92 | 74.31 | 75.50 | 76.23 | 69.99 | 73.99 |
| SSKD | 75.64 | 75.72 | 71.34 | 73.71 | 74.88 | 76.01 | 78.53 | 71.91 | 74.72 |
| ReviewKD | 75.41 | 76.42 | 72.04 | 74.10 | 75.03 | 75.91 | 78.02 | 70.21 | 74.64 |
| DKD | 75.02 | 76.44 | 72.09 | 74.39 | 74.91 | 76.49 | 76.58 | 70.51 | 74.56 |
| CTKD | 74.89 | 76.20 | 71.98 | 74.31 | 74.99 | 76.37 | 76.98 | 70.89 | 74.58 |
| ML-LD | 74.89 | 76.45 | 71.64 | 73.85 | 74.68 | 75.60 | 76.88 | 70.79 | 74.35 |
| **Ours** | **76.79** | **77.09** | **72.77** | **75.18** | **75.97** | **77.94** | **79.27** | **72.52** | **75.94** |

**Table 2: Top-1 and Top-5 accuracy (%) comparisons of SOTA distillation methods on ImageNet. Part of the compared results are from [14]. From left to right, the methods are ordered from oldest to newest.**

| Acc. | Teacher | Student | KD | AT | RKD | CRD | SSKD | ReviewKD | DKD | CTKD | ML-LD | Ours |
|---|---|---|---|---|---|---|---|---|---|---|---|---|
| Top-1 | 73.31 | 69.75 | 70.66 | 70.70 | 71.34 | 71.38 | 71.41 | 71.61 | 71.70 | 71.51 | 71.28 | **72.03** |
| Top-5 | 91.42 | 89.07 | 89.88 | 90.00 | 90.37 | 90.49 | 90.44 | 90.51 | 90.41 | 90.47 | 90.15 | **90.69** |

relationship across multiple stages between samples for knowledge distillation. Note that the student's accuracy surpasses the teacher's in certain identical architecture pairs, such as WRN40-2→WRN40-1. This underscores our method's capability to comprehensively extract more valuable information from teachers.

**Results on ImageNet.** We further evaluated a teacher-student pair on the large-scale ImageNet and its downstream task, using ResNet34 as a teacher and ResNet18 as a student. As shown in Table 2, our MDR delivers the best accuracy in both Top-1 and Top-5 categories. Specifically, MDR improves the accuracy by 0.62% over SSKD for Top-1 accuracy. The accuracy improvement on ImageNet is less pronounced than on CIFAR100. This can be attributed to higher class similarity within categories on ImageNet, which is challenging for the SM training method. Despite this, MDR still achieves one of the highest incremental gains over existing methods, reducing the student-teacher accuracy gap to 1.28% (about 20% relative improvement) compared to 1.61% for the previous best. These results highlight MDR's remarkable effectiveness on large-scale datasets even in the presence of challenging class structures.

**Transferability of Learned Representations.** Beyond achieving superior accuracy on the object dataset, it is imperative for the student network to produce generalized feature representations that can exhibit robust transferability to novel semantic recognition datasets. To this end, we adopt the strategy of freezing the backbone $f^S(\cdot)$ that has been pre-trained on the upstream CIFAR-100. We then train two linear classifiers based on the fixed penultimate features for downstream classification on the STL-10 and Tiny-ImageNet, respectively [31]. Table 3 shows the ability of transfer learning using different KD methods. Specifically, our MDR method outperforms the best-competing DKD by an accuracy gain of 1.41% on STL-10 and an accuracy gain of 1.14% on TinyImageNet, demonstrating its superior transferability to various recognition tasks.

**Efficiency under Few-shot Scenario.** We evaluate our method against conventional KD, CRD, SSKD and CTKD in a few-shot learning environment, using retention rates of 25%, 50%, and 75% of the original training samples. To ensure a fair comparison, we maintain a consistent data split strategy for each few-shot scenario, while keeping the original test set intact. Our evaluation utilizes the

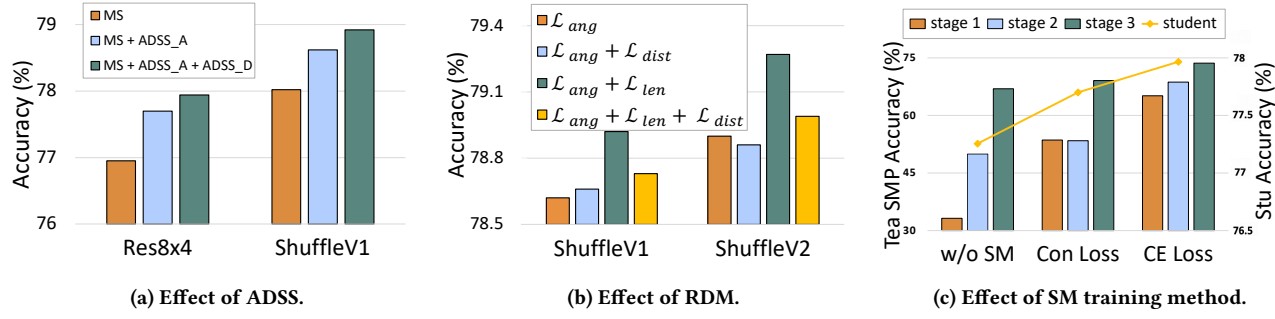

(a) Effect of ADSS.        (b) Effect of RDM.        (c) Effect of SM training method.

**Figure 3: Ablation study on CIFAR100. Student network ResNet8×4, ShuffleV1 and ShuffleV2, are trained under teacher network ResNet32×4.**

**Table 3: Linear classification accuracy (%) of transfer learning on the teacher-student pair ResNet32×4 → ResNet8×4.**

| Transferred Dataset | Baseline | KD | FitNet | RKD | CRD | SSKD | ReviewKD | DKD | Ours |
|---|---|---|---|---|---|---|---|---|---|
| CIFAR100→SIL-10 | 69.76 | 69.56 | 70.94 | 71.41 | 70.76 | 71.89 | 71.90 | 72.15 | **73.56** |
| CIFAR100→TinyImageNet | 34.29 | 34.77 | 38.07 | 38.02 | 38.17 | 38.56 | 38.54 | 38.74 | **39.88** |

**Table 4: Top-1 accuracy (%) comparison on CIFAR-100 under few-shot scenario with various percentages of samples.**

| Percentage | KD | CRD | SSKD | CTKD | Ours |
|---|---|---|---|---|---|
| 25% | 64.40 | 64.71 | 67.82 | 68.49 | **69.11** |
| 50% | 68.37 | 68.90 | 70.08 | 70.61 | **71.17** |
| 75% | 69.97 | 70.86 | 70.47 | 71.71 | **72.35** |

**Table 5: Comparison of detection mAP (%) on Pascal VOC using ResNet-18 as the backbone pre-trained on ImageNet by various KD methods.**

| Baseline | KD | CRD | SSKD | DKD | CTKD | Ours |
|---|---|---|---|---|---|---|
| 76.18 | 77.06 | 77.36 | 77.60 | 77.81 | 77.78 | **78.42** |

ResNet56-ResNet20 pair. As depicted in Table 4, our method consistently outperforms the other techniques by large margins across various few-shot scenarios. Notably, compared with the baseline trained on the complete set, our method achieves higher accuracy with only 25% of the training data. This outcome is attributed to our method's ability to effectively learn general relational information from limited data. In comparison, the previous methods typically focus on mimicking inductive biases from intermediate feature maps or incomplete relationships, which may overfit on the limited dataset and reduce generalization on the test set.

**Transferability for Object Detection.** We further evaluate the student network ResNet-18, which is pre-trained with the teacher

ResNet-34 on ImageNet, as a backbone for downstream object detection on Pascal VOC. For this evaluation, we adopt the Faster-RCNN [26] framework, adhering to the standard data pre-processing and fine-tuning protocols. Table 5 shows our method's superior detection performance, surpassing the original baseline by 2.24% mAP and the best-competing DKD method by 0.61% mAP. These results underscore our method's efficacy in guiding a network to achieve superior feature representations for diverse semantic tasks.

### 4.3 Ablation Studies

In this section, we provide ablation studies to analyze the effects of each component of MDR. The experiments are conducted on CIFAR-100 for classification task.

**Effect of Adaptive Stage Selection.** As shown in Fig. 3a, MS means using multi-stage decoupled relational information to transfer, which contains angle-wise and length-wise information in each stage. Applying angle-wise adaptive stage selection strategy (MS + ADSS_A) substantially boosts the accuracy upon the original multi-stage information, indicating that we extract a larger amount of beneficial angle-wise relationship. As we further add distance-wise adaptive stage selection strategy (MS + ADSS_A + ADSS_D), an even higher accuracy is achieved thanks to the positive contribution of valuable distance-wise information.

**Effect of Relational Decoupled Module.** To explore the effectiveness of proposed RDM, we conduct the evaluation in three variants: only using angle-wise information ($\mathcal{L}_{ang}$), angle-wise and distance-wise information ($\mathcal{L}_{ang} + \mathcal{L}_{dist}$), both angle-wise and length-wise information ($\mathcal{L}_{ang} + \mathcal{L}_{len}$) and all three information ($\mathcal{L}_{ang} + \mathcal{L}_{len} + \mathcal{L}_{dist}$). The results are shown in Fig. 3b, where coupled information ($\mathcal{L}_{dist}$) often have a negative impact on distillation, and RDM boosts the accuracy compared to the others.

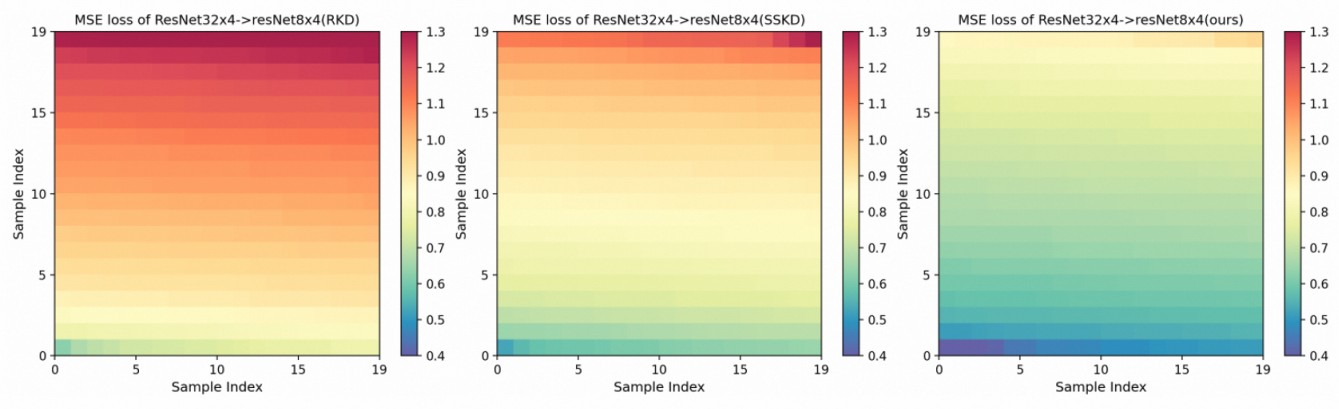

**Figure 4: MSE loss of relational matrix between ResNet32×4 and ResNet8×4. We visualize the MSE loss of relational matrix between the models trained by RKD (left), SSKD (middle), and MDR (right).**

**Table 6: Ablations on the number of adaptive stages.**

| Tea-Stu pair | info type | N = 1 | N = 2 | N = 3 |
|---|---|---|---|---|
| VGG13→VGG8 | angle | **75.97** | 75.88 | 75.83 |
| VGG13→VGG8 | length | **75.97** | 75.81 | 75.84 |
| Res50→MobV2 | angle | **72.52** | 72.32 | 72.19 |
| Res50→MobV2 | length | **72.52** | 72.29 | 72.33 |

To assess the impact of CE loss for SM training, we compare the teacher's SMP accuracy and student's accuracy under three cases: no SM, with contrastive loss, and with CE loss. SMP accuracy shows the faction of positive samples correctly assigned to the corresponding anchor. As shown in Fig. 3c, compared with the no-SM case, training with contrastive loss improves SMP accuracy, which is further improved by CE loss in both types of accuracy. Among them, the improvement of SMP accuracy in the early stage is more obvious with the modification of SM training strategy.

**The number of adaptive stages.** We validate various number of adaptive stages based on angle and distance respectively: 1/2/3 with two teacher-student pairs, including identical and distinct architectures. For the fairness of the comparison, when verifying the number of adaptive stages based on angle, length-wise one is fixed to be 1, and vice versa. As shown in Table 6, regardless of angle-wise or length-wise information, the best result is achieved when the number of adaptive stages is 1. Combined with stage selection, the accuracy improves steadily.

Due to the page limitation, more ablation studies and experiment analysis can be found in the supplementary materials.

## 4.4 Visualizations

In this part, we present the visualization to show that our MDR does bridge the teacher-student gap in the relation-level. The experiments are conducted on the sampled CIFAR-100 validation set (10,000 samples). We compute relational matrix with a batch size of 25 for the penultimate stage, so these are 400 values for

each experiment. We visualize the MSE loss of relational matrix between ResNet32×4 and ResNet8×4 in Fig. 4, which is formed by the addition of normalized angle-wise and length-wise matrix. For better presentation, we rank these values and organize them as the heatmap representation. The smaller the value, the more similar the matrix are. We can find that our MDR significantly improves the similarity of angle-wise and length-wise relational matrix between the student and the teacher.

Due to the page limitation, more related visualizations can be found in the supplementary materials.

## 4.5 Limitations

Compared with other relational distillation methods that only use the penultimate layer information for distillation, our method needs to use the middle layer information, so the teacher and the student need to have the same number of stages. Therefore, there is a constraint on the selection of distillable networks.

In addition, similar to other knowledge distillation methods based on contrastive learning, our method needs to first train the teacher's SM module and obtain the relationship matrix of angle and length, so it takes a longer time than the traditional KD method (under the same hardware conditions , the training time is 5.2 times that of traditional KD and 1.1 times that of SSKD).

## 5 CONCLUSION

In this paper, we propose a novel framework equipped with an adaptive stage selection strategy for relation-based knowledge distillation, which enables efficient extraction of relational information across multiple stages. By decoupling the relationship into angle and length difference and introducing a novel training method for the self-supervised module, our approach enables the student to acquire knowledge more effectively. Experiment results show that our method significantly surpasses SOTA performance on the standard image classification benchmarks in the field of KD. It also opens the door for further improvements of knowledge transfer methods based on relationship.

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
