# OpenReview forum: "MDR: Multi-stage Decoupled Relational Knowledge Distillation with Adaptive Stage Selection"
_acmmm.org/ACMMM/2024/Conference — MM2024 Poster_

### Official Review · Reviewer_5qsq · 2024-04-29

**Rating:** 4
**Confidence:** 3

**Summary:**

This paper introduces a multi-stage knowledge distillation method based on contrastive learning.  The proposed method contains two main folds, using the intermediate feature for feature distillation, and decoupling feature relationships to length and angle distances. Experiments demonstrate the efficacy of the proposed method.

**Strengths:**

1. The paper is well-written and easy to follow.
2. The proposed method is easy to apply and straightforward.
3. The authors conducted abundant experiments not only on classification but detection as well.

**Limitations:**

1. Did the authors reimplement the SOTA methods? They look different from the original papers.
2. Why do the authors use top-K to choose the layer rather than bottom-K or other methods? What is the rationale behind the design?
3. In Fig.3 (b), the result of L_len is missing.

**Suitability:**

2

---

### Official Review · Reviewer_wBch · 2024-05-24

**Rating:** 3
**Confidence:** 2

**Summary:**

The article discusses a novel framework for relation-based knowledge distillation (KD) that incorporates an adaptive stage selection strategy to efficiently extract relational information across multiple stages. This method decouples the relationship between samples into angle and length differences, enhancing the transfer of relational knowledge from teacher to student models. The proposed approach significantly improves performance on standard image classification benchmarks and addresses limitations in existing contrastive-learning-based methods by extracting effective relational information from intermediate layers.

**Strengths:**

The article introduces a novel framework for relation-based knowledge distillation that incorporates an adaptive stage selection strategy and relationship decoupling into angle-wise and length-wise components. This approach enhances the efficiency of relational information extraction across multiple stages, leading to more effective knowledge transfer. The proposed method significantly outperforms state-of-the-art techniques in image classification benchmarks, demonstrating superior accuracy and potential for further improvements in knowledge distillation methods.

**Limitations:**

1.Limited Comparison: The paper lacks a comprehensive comparison with a wider range of existing knowledge distillation methods, potentially restricting the assessment of MDR's performance against a broader set of benchmarks.
2.Scalability Concerns: While MDR demonstrates effectiveness in certain tasks, the scalability of the method to larger and more complex models or datasets has not been thoroughly explored, leading to questions about its applicability in diverse scenarios.
3.Generalization: The discussion on the generalizability of MDR across different domains or applications is limited, creating uncertainty regarding its effectiveness in varied contexts.
4.Theoretical Depth: The paper could benefit from more in-depth theoretical analysis or comparisons with existing theoretical frameworks in knowledge distillation to enhance the academic rigor of the proposed MDR approach.

**Suitability:**

2

---

### Official Review · Reviewer_4Wr4 · 2024-05-26

**Rating:** 4
**Confidence:** 4

**Summary:**

This paper introduces a novel KD framework that enhances distillation by using multi-stage relational information. It decouples angle-wise and distance-wise information to improve the transfer of relational knowledge and resolves conflicts between these attributes. The framework features an Adaptive Stage Selection (ADSS) strategy to identify the most effective stages for KD. Extensive experiments on CIFAR-100, ImageNet, and Pascal VOC demonstrate MDR's superior performance, surpassing state-of-the-art methods with an average improvement of 1.22% on CIFAR-100.

**Strengths:**

1. This paper proposed an adaptive stage selection strategy to decouple the relational information.  This innovative perspective addresses previously unrecognized conflicts in relational information for knowledge distillation.
2. The structure of this paper is logical and coherent, beginning with an introduction to the problem, followed by a detailed explanation of the proposed methodology, and concluding with experimental results and discussions.

**Limitations:**

1. Some sections explaining complex concepts like angle-wise and length-wise relational information are difficult to understand and would benefit from more diagrams and simplified explanations.
2. The method requires the teacher and student networks to have the same number of stages, limiting its applicability to certain network architectures.
3. It is recommended that the illustrations in the introduction should not display the method results, but rather showcase novel findings that explain the proposed problems and phenomena.

**Suitability:**

2

---

### Official Review · Reviewer_iXed · 2024-05-27

**Rating:** 4
**Confidence:** 3

**Summary:**

This paper propose a Multi-stage Decoupled Relational (MDR) KD framework equipped with an adaptive stage selection which enables multi-stageinformation extraction to maximize the efficacy of transferring the relational knowledge. MDR framework decouples angle-wise and distance-wise information to resolve their conflicts while still preserving complete relational knowledge by resulting in an elevated transferring efficiency and distillation quality.This paper also formulate a new SM trainingparadigm to compensate for the loss of length-wise information and augment its contrastive learning representation capability.

**Strengths:**

1. The sentence structure of the paper is fluent and there are no obvious grammar or spelling errors.
2. The paper has conducted rich comparative experiments and ablation experiments, as well as supplementary materials for ablation experiments and experimental analysis, which provide strong support for the experimental results.
3. The paper provides a novel multi-stage distillation framework with adaptive stage selection strategy, which extracts effective relationship information from the intermediate layer and is innovative.
The method proposed in the paper has good performance in tasks such as classification, few shot learning, transfer learning, and object detection, and has good generalization ability.

**Limitations:**

1. The process in Figure 2 needs to be explained more clearly, for example, what standards are used for information filtering in areas with red crosses in the figure? I hope it can be reflected in the picture.
2. In the experimental setup of section 4.1, there was no detailed introduction to the data augmentation methods used in the experiment. Only line 299 mentioned the use of methods such as Random Rotation and MixUp. Please provide a specific list of the data augmentation methods used in the experiment.
3. The author validated and compared the performance of the method across multiple datasets, but the comparison on ImageNet is relatively weak, which should ideally be the most crucial comparison. It is recommended that the author strengthens this aspect of the content to increase the reliability of the paper.
4. This study appears to be quite interesting and shows some degree of innovation. It is suggested that the author better elucidate the motivation behind this paper to enhance its readability, aid in understanding the author's work, and assist readers in evaluating the performance of this study.

**Suitability:**

2

---

### Meta-Review · Area_Chair_oGvu · 2024-06-30

**Recommendation:** Accept (Poster)
**Confidence:** 4

**Metareview:**

This paper proposes a Multi-stage Decoupled Relational (MDR) KD framework with adaptive stage selection, which allows for effective multi-stage information extraction to enhance relational knowledge transfer. The MDR framework separates angle-wise and distance-wise information to avoid conflicts, ensuring the preservation of complete relational knowledge and improving both transfer efficiency and distillation quality. Extensive experiments across various benchmarks and tasks validate the effectiveness of the method. Although the applicability of MDR to network architectures with different stages remains unverified (pointed out by Reviewer 4Wr4), most of the concerns are well-addressed during the rebuttal and most reviewers acknowledge the novelty and the effectiveness of the proposed method. Therefore, the AC recommends to accept this submission.